# Not all who wander are lost: Trail bias in community science

**Ellyne M. Geurts** [1][0]*, **John D. Reynolds**[2][0], **Brian M. Starzomski**[1][0]*

**1** School of Environmental Studies, University of Victoria, Victoria, British Columbia, Canada, **2** Earth to Ocean Research Group, Department of Biological Sciences, 8888 University Drive, Simon Fraser University, Burnaby, British Columbia, Canada

[0] These authors contributed equally to this work.
* egeurts@uvic.ca (EMG); starzom@uvic.ca (BMS)

**Data Availability Statement:** Data and code used in this study can be found on GitHub (https://github.com/ellynegeurts/Fine_scale_spatial_biases_iNaturalist_British_Columbia). This information can also be accessed on Zenodo (DOI: 10.5281/zenodo.7761835).

## Abstract

The exponential growth and interest in community science programs is producing staggering amounts of biodiversity data across broad temporal and spatial scales. Large community science datasets such as iNaturalist and eBird are allowing ecologists and conservation biologists to answer novel questions that were not possible before. However, the opportunistic nature of many of these enormous datasets leads to biases. Spatial bias is a common problem, where observations are biased towards points of access like roads and trails. iNaturalist–a popular biodiversity community science platform–exhibits strong spatial biases, but it is unclear how these biases affect the quality of biodiversity data collected. Thus, we tested whether fine-scale spatial bias due to sampling from trails affects taxonomic richness estimates. We compared timed transects with experienced iNaturalist observers on and off trails in British Columbia, Canada. Using generalized linear mixed models, we found higher overall taxonomic richness on trails than off trails. In addition, we found more exotic as well as native taxa on trails than off trails. There was no difference between on and off trail observations for species that are rarely observed. Thus, fine-scale spatial bias from trails does not reduce the quality of biodiversity measurements, a promising result for those interested in using iNaturalist data for research and conservation management.

## Introduction

Community science, also called citizen science, involves data collected in collaboration between community members and scientists. It is growing rapidly in popularity in biodiversity research and management [1–3]. The advancements of technology and accessible platforms like eBird and iNaturalist have increased participation and are producing massive amounts of biodiversity and ecological data around the world [4, 5]. These data have been used to answer questions of phenology [6, 7], species range changes [8], species migrations [9], phenotypic variation [10, 11], species interactions [12], habitat use [13, 14], and body condition and disease [15]. The data from these platforms are also helping researchers discover new species and track invasive species [16, 17], improve species distribution models [18, 19], and increase our understanding of large scale effects of climate on animal behaviour [20]. In addition to

**Funding:** This work was supported by BC Parks (https://bcparks.ca/), the BC Ministry of Forests, Land, Resource Operations and Rural Development (now known as the BC Ministry of Water, Land and Resource Stewardship: https://www2.gov.bc.ca/gov/content/governments/organizational-structure/ministries-organizations/ministries/water-land-and-resource-stewardship), the Sitka Foundation (https://sitkafoundation.org/), the Pacific Wildlife Foundation (https://pwlf.ca/), and the Natural Sciences and Engineering Research Council of Canada (NSERC: https://www.nserc-crsng.gc.ca/index_eng.asp). Provincial funding (BC Parks and the BC Ministry of Forests, Land, Resource Operations and Rural Development) was received by B.M.S. Funding from the Sitka Foundation (Grant #: 31-R831693) and the Pacific WildLife Foundation was received by J.D.R. In addition, NSERC Discovery grants were received by B.M.S. (Grant #: 04476) and J.D.R. (Grant #: 31-R611506). Lastly, a NSERC CGS-M scholarship was received by E.M.G. The funders had no role in study design, data collection and analysis, decision to publish, or preparation of the manuscript.

**Competing interests:** The authors have declared that no competing interests exist.

contributing large amounts of ecological data, community science platforms are educational resources that connect the public with their environments and with other like-minded individuals, cultivating a community of naturalists and environmental stewards [1, 21, 22]. As a result, researchers and environmental managers are keen to promote these platforms to collect more and better data [23–25].

iNaturalist engages the largest number of people globally of any biodiversity community science platform, with more than 138 million observations–geotagged photographed or audio-recorded species records–of taxa around the world, serving as a huge voucher photo and audio data repository [3]. The photo vouchers provide primary data on species occurrences, but also valuable secondary data such as life stage and sex, species interactions, and body condition. Users of iNaturalist are self-directed (*i.e.*, no sampling guidelines) and can make a species observation anywhere in the world at any time. This open-geographic feature allows users to contribute data across broad geographic scales that could never be obtained from traditional scientific surveys [26]. Nevertheless, reservations remain regarding the quality of the data from iNaturalist due to its lack of sampling guidelines, which may lead to spatial, temporal, and taxonomic biases (Di Cecco et al., 2021).

Virtually all biodiversity datasets suffer from biases on spatial, temporal, taxonomic, and observer levels [27–31]. Community science data are frequently biased spatially by level of accessibility, such as proximity to roads and trails [32–34]. Observations are also biased towards tourist locations, such as parks and protected areas [35, 36], particular habitats [37–39], and by human population density [29, 31, 40]. These biases need to be understood and properly accounted for [41–43]. Some observational studies of the spatial, taxonomic, observer, and temporal biases in iNaturalist data have been addressed [37], but no experimental studies of iNaturalist biases and their impacts on biodiversity records have been conducted.

To fill this gap, we tested fine-scale spatial impacts on taxonomic richness estimates. We compared observations made along trails and away from trails, because in many terrestrial habitats observations are biased toward trails due to accessibility [32, 44, 45]. This bias may be reinforced by rules in parks, where people are not allowed to leave trails [46]. We asked the question "Does trail spatial bias affect the number of taxa observed in parks by community scientists?". We conducted timed field experiments using experienced iNaturalist observers (2–5 years using the platform) to compare taxonomic richness estimates between on-trail versus off-trail observations in provincial parks in British Columbia, Canada. The experiments emulated how an iNaturalist user would observe their environment with no distance or taxonomic restriction for each transect. We predicted higher taxonomic richness estimates on-trail compared to off-trail based on the intermediate disturbance hypothesis [47–49]. In addition, we predicted the species compositions to vary between on and off trail, with more exotic species observed on-trails [50]. Lastly, we examined differences in rare and locally threatened species observations between on and off trail.

## Materials and methods

### Study area and data

Sampling took place in 22 provincial parks and protected areas in British Columbia, Canada (S1 Fig). We conducted transects in three habitat types: grasslands, open-canopy forests, and closed-canopy forests (Fig 1, S1 Table). Grasslands lack trees, open-canopy forests are dominated by pine trees (*Pinus* sp.), and closed-canopy forests are dominated by non-pine trees. We surveyed a variety of trail substrate types: bare ground (n = 63), gravel (n = 33), and asphalt (n = 1). Trail widths ranged from 0.5 m to 10 m with a mean of 3.1 m. Note that the 10 m trail was a groomed cross-country trail.

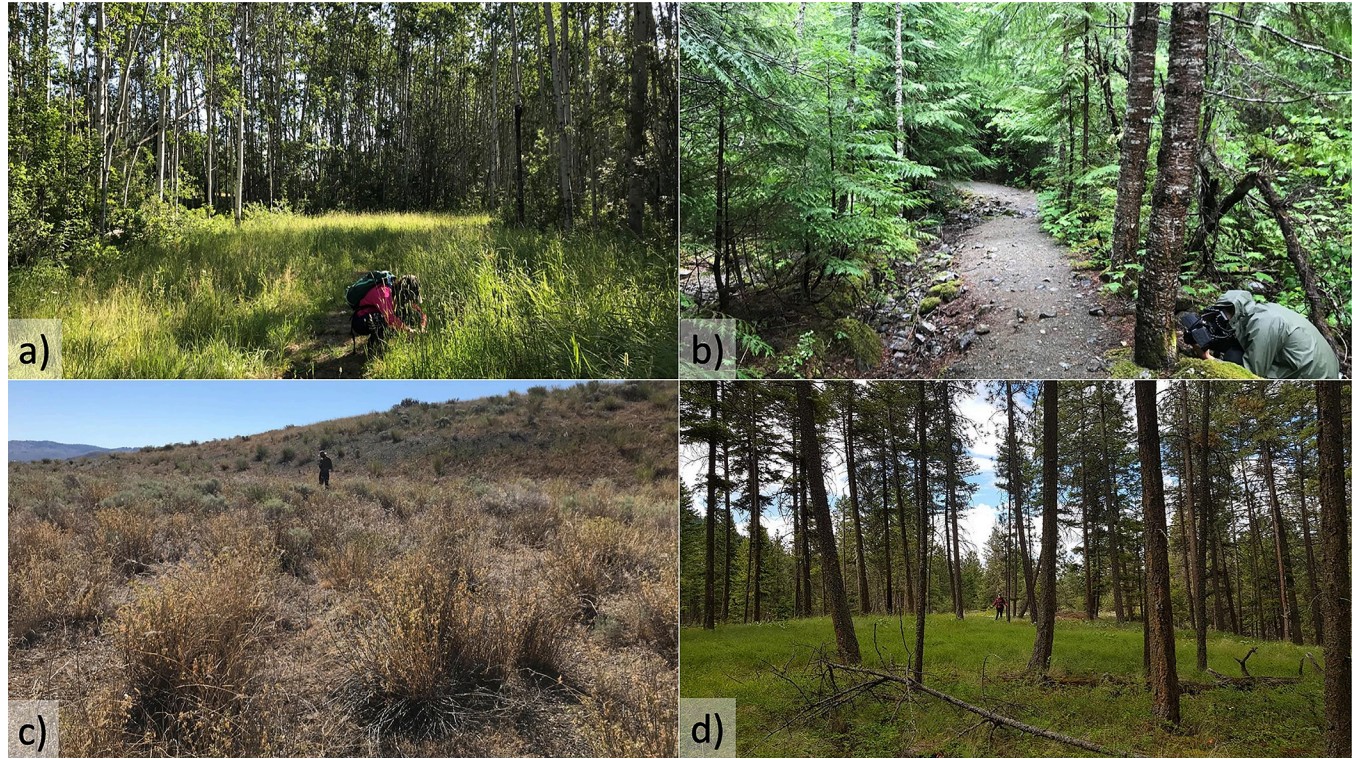

**Fig 1. Examples of habitats surveyed in British Columbia, Canada. a)** Closed-canopy forest (Beatton Provincial Park). **b)** Closed-canopy forest (Nairn Falls Provincial Park). **c)** Grassland (Steelhead Provincial Park). **d)** Open-canopy forest (Ellison Provincial Park). Photos taken by Kate McKeown (a-c) and Ellyne Geurts (d). Published with permission from Kate McKeown, original copyright 2021.

We defined taxonomic richness as the number of unique taxa per transect. Taxonomic richness includes all taxonomic groups (*e.g.*, plants, animals, fungi). Unique taxa are different organisms observed that may not be identified to species-level. For example, dandelion (*Taraxacum* sp.), darkling beetle (Tenebrionidae), and a lichen (Lecanoromycetes) would be counted as three taxa. This allowed for higher classification level observations to be included, as not all observations (*e.g.*, spiders and lichens) could be identified to species level by photographs alone [51, 52]. We analyzed the number of exotic, locally threatened, and rare species detected on and off trails. Exotic species are species that are established outside of their native range via anthropogenic means. The list of species not-native to British Columbia was downloaded from the BC Species & Ecosystems Explorer from the Conservation Data Centre [53]. Locally threatened species have a status in British Columbia that ranges from "special concern" (S3) to "historical species or possibly extirpated communities" (SH). The full list can be found on the "BC rarities project" on iNaturalist.ca (https://inaturalist.ca/projects/bc-rarities?tab= about –accessed January 13, 2022). We define rare species as species that have only one record on iNaturalist in the park where the transects were done. We downloaded the transect data from iNaturalist using its CSV Export Function on November 18, 2021 (n = 9,931 observations), and downloaded all observations recorded in the 22 parks on January 30, 2023 (n = 94,777 observations).

## iNaturalist

iNaturalist is a crowd-source community science platform that was founded in 2008 and has since grown to over 2.6 million users globally [54]. It is hosted by the California Academy of

Sciences and the National Geographic Society. Users can upload georeferenced photo and audio recordings of any wild organism (*i.e.*, a verifiable observation). Photo identification can be aided by iNaturalist's computer vision program as well as input from the iNaturalist community. If the verifiable observation reaches majority agreement in species identification from the community, the quality grade of the observation is upgraded from "Needs ID" to "Research Grade", and depending on license restrictions by the observer, the data are incorporated in the Global Biodiversity Information Facility (https://www.inaturalist.org/pages/help). iNaturalist produces opportunistic presence-only data.

## Field experiments

We conducted field experiments of trail spatial biases with a team of four iNaturalist observers from the BC Parks iNaturalist Program (https://inaturalist.ca/projects/bc-parks and https://inaturalist.ca/projects/bc-parks-inat-team-big-summer-2021) [55]. The observers were active generalist users with at least 3,500+ observations on iNaturalist and had prior experience surveying biodiversity in British Columbia, such as plants, invertebrates, vertebrates, fungi, and lichens. Observers did surveys together for the first two weeks with professional naturalists before beginning the field experiments to help standardize identification and search patterns. Data collection took place between May and August 2021 (S1 Table). We conducted 96 paired timed transects with no distance restriction. We photographed the habitat at the beginning and end of each transect to help refine habitat categories. We conducted the paired transects in teams of three, with two observers and one facilitator for efficient data recording. One observer walked on the trail while the other observer walked parallel to the trail approximately 20 m away within the same habitat. We selected 20 m because trail effects from disturbance generally fade after 5 m away from the trail [49, 56]. The facilitator followed behind the observer pair to keep time, gather all metadata for each transect, and ensure the off-trail observer stayed at least 20 m from the trail. We implemented a duration limit to aid comparability between the two treatments. The observer pairs photographed all plants, animals, and fungi that they encountered for 30 minutes in each transect. They were instructed simply to try to maximize the number of species that they photographed. They recorded their straight-line distances using Garmin GPS units (*etrex 20*) as a proxy for distance traveled. We did not measure true distance traveled due to GPS signal interferences in forests and ravines resulting in unreliable GPS track distances. Transects did not have set distances to better emulate a typical iNaturalist user, but we considered the effects of distances traveled in the statistics. The on-trail transects included a 2 m buffer on either side of the trail to mimic typical iNaturalist user behaviour in photographing species from trails. We selected trail segments that did not contain switchbacks to allow parallel surveying without backtracking. The side of the off-trail survey was selected randomly except when there were safety concerns.

During the on and off trail transects, the facilitator recorded trail width and type, and the dominant habitat type and mesohabitats that the observers encountered [57, 58]. Mesohabitats included cliff faces (large vertical rock), seeps (damp depressions), streams (narrow flowing water), and water pools (small areas with standing water). The observers employed the floristic habitat sampling method also referred to as the "intelligent meander" [59], where observers seek different mesohabitats and microhabitats (*e.g.*, logs and rocks) during transects. This sampling method detects more species, especially rarer species, than the traditional plot sampling method and is thought to more closely resemble the behaviour of iNaturalist users [57]. To ensure a balanced sampling design, each observer surveyed on and off trail an equal number of times throughout the summer. In addition, observer pairs switched regularly. Each observer surveyed with each partner the same number of times. To avoid temporal clustering of

sampling for both trail position and observer partner, the rotations were spaced out evenly throughout the summer. Surveys were done opportunistically throughout the summer when parks with adequate trails were available and weather permitted. No permits were required to conduct this field study in the provincial parks as no specimens were collected.

## Statistical analyses

We compared on versus off trail taxonomic richness using generalized linear mixed models with a Poisson distribution using the "glmer" function from the *lme4* R package [60]. Taxonomic richness is defined as the number of unique taxa recorded per transect. We included observer identity as a random effect and the trail variable of on- and off- position as a fixed effect. We also included location of transect as a random effect with trail name nested within park name. In addition, we considered habitat type, number of observations, and distance traveled as additional explanatory variables. We standardized the distance traveled variable to have a mean of zero and standard deviation of one. We excluded the variable of mesohabitats encountered as most of the transects encountered zero mesohabitats (n = 168 out of 192 individual transects), meaning there was only one dominant habitat type throughout the transect. We used random intercepts for the variance structure. We did not test for random slopes for different observers due to high ratio of parameters to sample size. We compared four models with different fixed-effect structures using AICc: (1) a full model with trail position (on vs. off), habitat type, and distance traveled, (2) a full model with an interaction term for distance traveled and trail position, as distance traveled could be affected by whether the observer is on or off trail, (3) a model with only the trail position variable, and (4) a null model. We then repeated these steps with exotic species observations removed from the analysis (n = 391 observations removed) to compare native taxonomic richness.

We compared the number of rare species observed on or off trails using a Wilcoxon sign rank test. We began by filtering all observations in each park to only contain species-level observations (n = 57,843), then we removed all "casual" quality grade observations from the dataset (n = 57,547). For each park we created individualized rare species lists by selecting species that only had one record in the park. This allowed us to count the number of rare species observations recorded by us on and off trail by park. We treated each park as a pair (n = 22). We also counted the number of locally threatened species recorded on and off trails across all parks. Similarly, we compared the number of exotic species observed between on and off trail using a Wilcoxon sign rank test. We grouped transects by park and summed the number of exotic species observations recorded on and off trail. We also calculated the proportion of observations in each transect that contained exotic species.

We conducted the statistical analyses and data preparation in R (version 4.1.2.) and RStudio [61, 62]. The package we used for statistical analyses was *lme4* [60], and for data cleaning, management, and visualization we used *lubridate* [63], *dplyr* [64], *bbmle* [65], *sp* [66], *rgdal* [67], *adehabitatLT* [68], and *ggplot2* [69]. We archived the data and statistical analysis code we used on Zenodo (DOI: 10.5281/zenodo.7761835).

## Results

### Trail bias–taxonomic richness estimates

The observers recorded a total of 9,931 observations (on-trail = 5,107, off-trail = 4,824) of 1,323 taxa across the 96 paired transects. As predicted, there was higher taxonomic richness on trails than off trails. The mean number of taxa on-trail was 40.8, while off-trail was 37.6.

The best model was the full model with all variables included and no interaction between trail position and distance traveled (Table 1). Both the trail position and habitat variables were

**Table 1. Models tested to explain taxonomic richness observed along transects.**

| Model name | Variables tested | ΔAICc | DF | Weight |
|---|---|---|---|---|
| *Total taxonomic richness* | | | | |
| **Full model** | Trail position + distance + habitat | 0 | 8 | 0.671 |
| **Full model with distance interaction** | Trail position + distance + habitat + trail position:distance | 2.1 | 9 | 0.231 |
| **Trail position only** | Trail position | 3.9 | 5 | 0.097 |
| **Null model** | 1 | 15.1 | 4 | < 0.001 |
| *Native taxonomic richness* | | | | |
| **Full model** | Trail position + distance + habitat | 0 | 8 | 0.729 |
| **Full model with distance interaction** | Trail position + distance + habitat + trail position:distance | 2.1 | 9 | 0.256 |
| **Trail position only** | Trail position | 8.3 | 5 | 0.012 |
| **Null model** | 1 | 10.4 | 4 | 0.004 |

significant in that model (Fig 2). The grassland habitat had a lower number of observed taxa than the closed-canopy forest habitat (Figs 2 and 3). The number of taxa observed was highly correlated with number of observations (Pearson's product-moment correlation: r = 0.90, t = 29.27, df = 190, p < 0.001). Distance traveled did not affect the number of taxa observed (Fig 2). Mean and standard error of distance traveled by observers on-trail was 114 ± 9 m and 89 ± 6 m for off-trail.

When we restricted the analysis to native taxa, the same top model explained taxonomic richness (Table 1). The pattern of variable importance is also similar, except 'open-canopy forest' now has an effect (Fig 2).

Total taxonomic richness is the total number of unique taxa observed per transect. Native taxonomic richness is total taxonomic richness with exotic species removed (n = 391

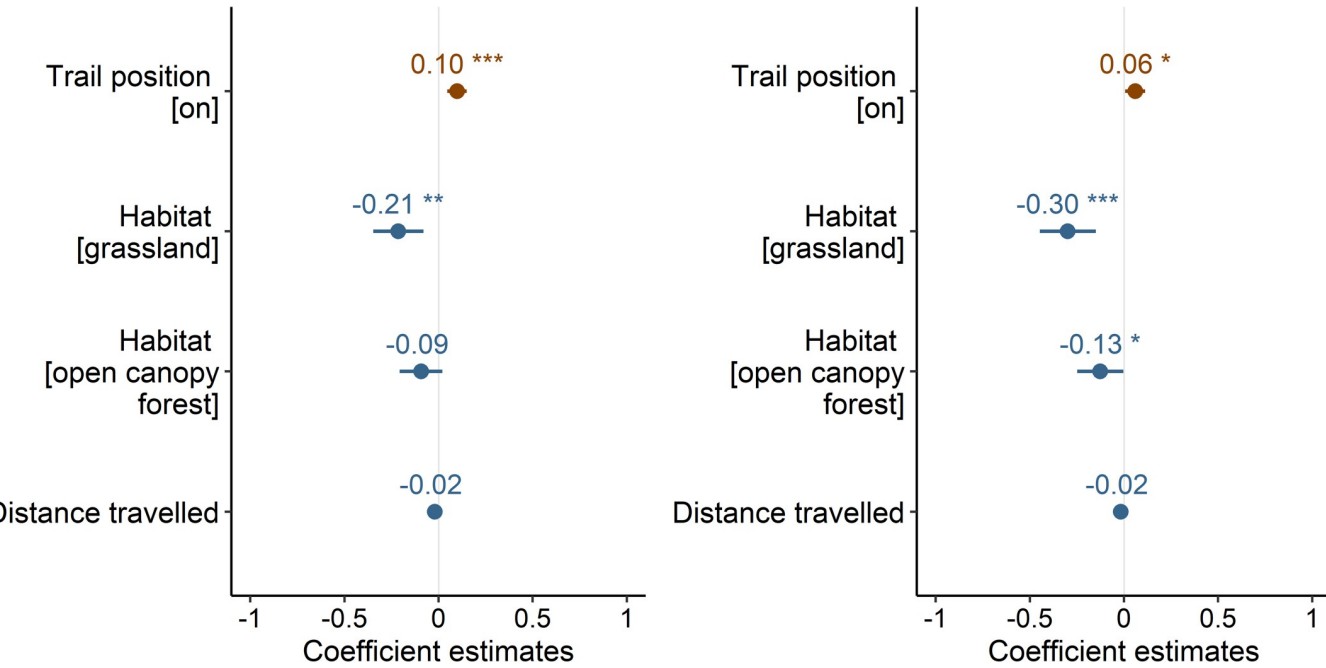

**Fig 2. Standardized coefficient estimates with standard error bars for variables in the top model examining taxonomic richness by trail position, habitat type, and straight-line distance traveled by observer.** Trail position is "on" or "off". The three habitat types were "grassland", "open-canopy forest" and "closed-canopy forest". * indicates p < 0.05, ** indicates p < 0.01, and *** indicates p < 0.001. Blue indicates a negative relationship between the covariate and taxonomic richness, while brown indicates a positive relationship. a) Total taxonomic richness model. b) Native taxonomic richness.

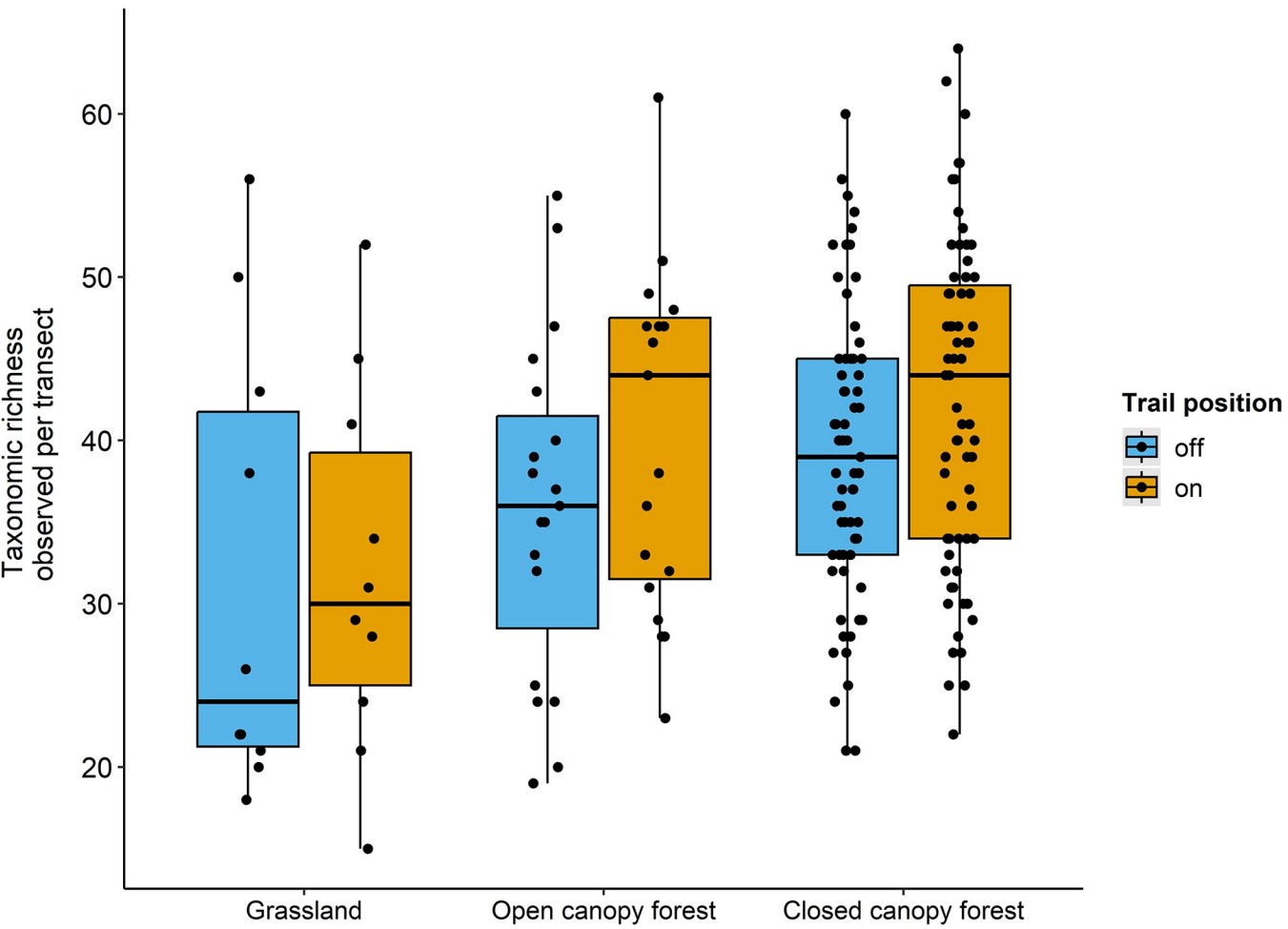

**Fig 3. Observed taxonomic richness per transect across the three habitat types.** Taxonomic richness is the number of unique taxa on iNaturalist per transect. Black bar = median. Box = interquartile range. See Fig 2 for the model coefficient results.

removed). Trail position indicates whether the observer was on or off trail. Distance = straight-line distance traveled during a transect. Habitat is classified into three broad categories reflecting general vegetation structure. All models had observer and trail name nested within park name as random effects.

### Trail bias–locally threatened, rare, and exotic species

Seven of the 192 individual transects contained at least one locally threatened species (on-trail = 6 transects, off-trail = 1 transect). Among the seven transects, only one transect had multiple threatened taxa (n = 2). Examples of locally threatened species are shown in Fig 4. We observed four insect species: Bunch Grass Locust (*Pseudopomala brachyptera*), a potter wasp (*Odynerus dilectus*), Kiowa Grasshopper (*Trachyrhachys kiowa*), and Huron Short-winged Locust (*Melanoplus huroni*). We also observed one bird species (White-throated Swift—*Aeronautes saxatalis*) and one plant species (Large-flowered Triteleia (*Triteleia grandiflora*). There was no difference in number of rare species between on and off trail (Wilcoxon signed rank test with continuity correction: V = 73.5, p = 0.40).

More than half of the individual transects (110 of the 192) had at least one exotic species (Table 2). As predicted, we observed more exotic species on than off trail (Wilcoxon signed

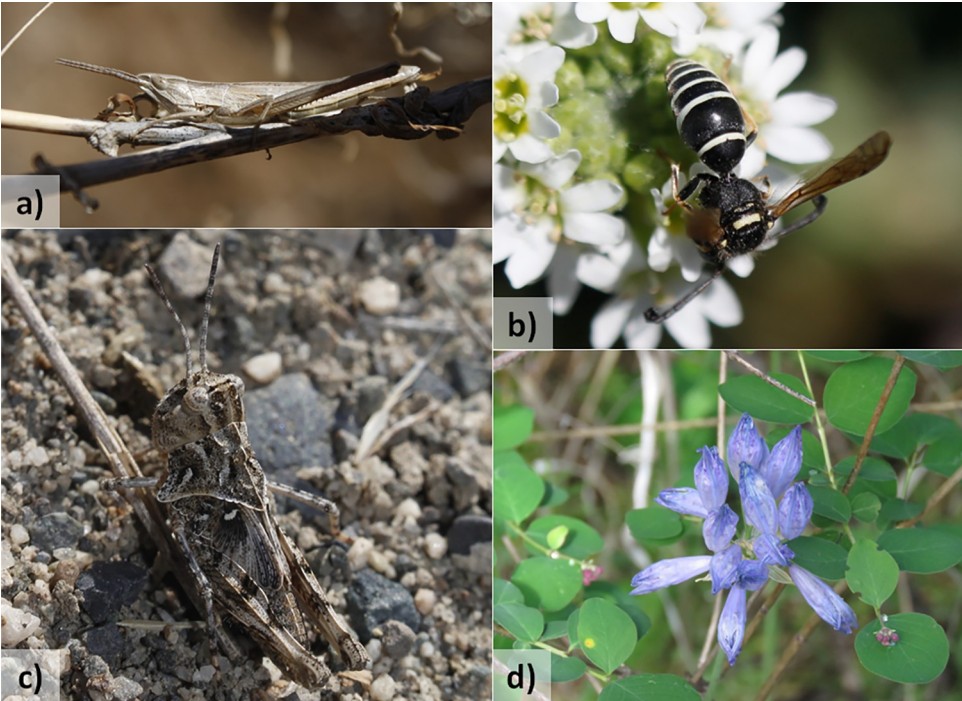

**Fig 4. Locally threatened species observed during transects on-trail in British Columbia, Canada.** Locally threatened species range from "special concern" (S3) to "historical species or possibly extirpated communities" (SH) in British Columbia. **a)** Bunch Grass Locust (*Pseudopomala brachyptera*) observed in Steelhead Provincial Park. **b)** *Odynerus dilectus* observed in Fintry Provincial Park and Protected Area. **c)** Kiowa Grasshopper (*Trachyrhachys kiowa*) observed in Steelhead Provincial Park. **d)** Large-flowered Triteleia (*Triteleia grandiflora*) observed in Kalamalka Lake Provincial Park. Photos taken by Lena Dietz Chiasson (a-c) and Erin Springinotic (d). Reprinted from iNaturalist under a CC BY license, with permission from Lena Dietz Chiasson and Erin Springinotic, original copyright 2021.

rank test with continuity correction: V = 2.5, p < 0.001). Exotic species appeared to make up a higher proportion of the observations for transects in grassland habitats than in forest habitats (Fig 5). See Table 2 for mean and range of number of exotic species observed on and off trails.

## Discussion

Higher taxonomic richness observed on trails than off trails during our 30-minute biodiversity transects suggests that fine-scale bias (*i.e.*, trail bias) does not lead to reduced biodiversity measurements from opportunistic community science. The lack of difference between on and off trail transects for rare (*i.e.*, uncommonly recorded) species also supports the inference that trail bias does not limit the quality of biodiversity observations derived strictly from trails. These results are encouraging evidence for park managers and conservation researchers: they show that spatially biased data from opportunistic community science can provide comparable or stronger taxonomic richness estimates on-trails compared to off-trails in a variety of habitats.

**Table 2. Summary statistics for the number of exotic observations and species recorded by trail position for all transects (n = 96 paired transects).**

| Trail position | Number of transects that encountered at least one exotic species | Mean number of exotic observations | Range of number of exotic observations | Mean number of exotic species | Range of number of exotic species |
|---|---|---|---|---|---|
| Off | 38 | 1.10 | 0–15 | 0.89 | 0–11 |
| On | 72 | 2.97 | 0–15 | 2.43 | 0–10 |

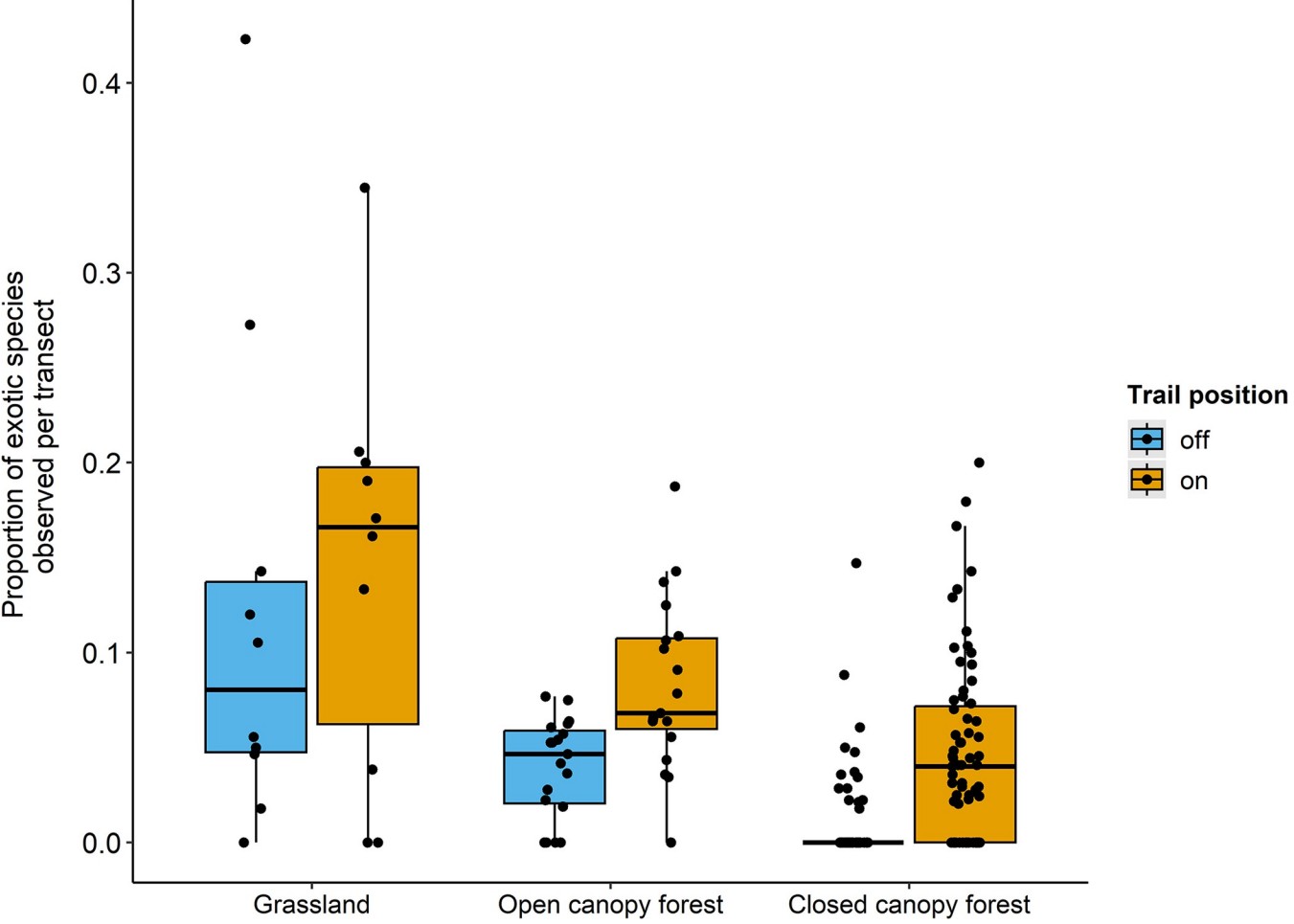

**Fig 5. The proportion of exotic species observed per transect across the three habitat types.** Proportion is calculated out of the total taxonomic richness observed per transect. Black bar = median. Box = interquartile range.

### Trail bias—taxonomic richness

Our test for effects of trails on estimates of taxonomic richness is novel because it includes all taxa in a variety of habitats in time-controlled transects, where observers conducted unstructured community science surveys. Previous results using standardized surveys have been mixed regarding trail impacts on species diversity [70]. Some studies have found higher species richness on trails for vascular plants and ground-dwelling arthropods [47–49, 71], while other studies found no difference *e.g.* plants [72] or even higher species richness away from trails *e. g.*, lichens, flower-visiting insects, and gall-inducing insects [72–75].

There are a few potential explanations for why we observed higher taxonomic richness on trail than off trails. It is possible that there were indeed more species along trails, perhaps because of more light and disturbance, which could favour ruderal species [48, 70, 71]. However, species may also have been more readily detected from trails due to increased visibility, less noise disturbance, and greater ease of observers moving through the habitat. Detectability differences between roadside and off-road surveys have been documented for birds, with roadside surveys having greater detectability of bird species when in high forest coverage regions than off-road surveys, and the opposite effect when forest coverage is low [76]. As the number

of taxa observed was highly correlated with number of observations, it is likely that observing from trails imparts advantages for biodiversity surveying, allowing for more species to be recorded than away from trails. Either way, if people want to see as many species as possible (and most naturalists do), the message is that they will do so by following trail etiquette.

## Trail bias—species composition differences

Our finding of more exotic species along trails than away from trails agrees with other studies [50, 77] and provides reassuring evidence for potential rapid detection of new invasives in regions [17, 78]. This result might misleadingly give the impression that the higher taxonomic richness on-trails versus off-trails is due to exotic species locally boosting trailside taxonomic richness. However, when we removed exotic species from the analyses, we still found evidence of increased taxonomic richness of native taxa on trails.

The similarity in detection of rare species on versus off trails also suggests that observing from trails does not impose a disadvantage on biodiversity measurements. In addition, we found more locally threatened species along trails than off trails, though our sample sizes were too small to make strong inferences. Most of the locally threatened species that we observed were winged *i.e.*, mobile species (Fig 4). It is thus possible that locally threatened plants and lichens could be less commonly observed along trails and consequently be underrepresented on iNaturalist [72].

Note that we used experienced generalist iNaturalist users to test for fine-scale spatial biases to reflect the ideal scenario of maximal biodiversity collection in a region as active users disproportionately contribute to biodiversity data on opportunistic community science platforms [5, 37, 79]. However, iNaturalist users exhibit a variety of sampling behaviours [37], consistent with other opportunistic community science projects, including variation in search images, detection abilities, equipment used, and taxonomic preferences [39, 80, 81]. Thus, our results using a small number of active observers does not capture the full observer variability present on the iNaturalist platform. The next step would be to repeat an experiment like this with inexperienced community scientists or use existing data on platforms like iNaturalist to analyze on versus off trail observations.

In summary, our finding of higher taxonomic richness observed along trails than away from trails by iNaturalist observers is a promising outcome for the value of the platform for surveying biodiversity. Of course, there are still going to be strong biases due to the lack of sampling guidelines on iNaturalist, and spatial coverage will be imperfect. For example, some microhabitats (*e.g.*, logs, rocks, stumps) and mesohabitats (*e.g.*, seeps, cliffs, wetlands) and their inhabitants may be underrepresented by observers who stay on trails. Targeted surveys would be needed to complement community science for species in such places [82]. In addition, the higher number of exotic species documented along trails and lack of difference detected for rare species supports the value of these data for tracking invasive species and for locating places where rare and locally threatened species occur. We hope that the results of more experiments like ours can further illuminate biases and enhance our ability to use the data effectively.

## Supporting information

**S1 Fig. The 22 provincial parks and protected areas where we conducted transects between May and August 2021, in British Columbia, Canada.** We used the ESRI Terrain and Reference Overlay basemaps from QGIS.
(TIF)

**S1 Table. Provincial parks and protected areas visited along with the number of paired transects conducted and the dominant vegetation encountered for each site.** Dominant vegetation types are the tallest trees and shrubs with the most cover. PP = Provincial Park. PA = Protected Area.
(PDF)

## Acknowledgments

We want to give a huge thank you to the field crew: Jason Headley, Erin Springinotic, Tori Miller, Kate McKeown, and Lena Dietz Chiasson for all their hard work and awesome observations. We also appreciate the 2,620 iNaturalist identifiers who volunteered their time and expertise to identify the observations we uploaded. The top 10 identifiers were Doug Brown (@dougbrown), Barbara L. Wilson (@sedgequeen), Randal Mindell (@rambryum), Finn McGhee (@fmcghee), Steve Ansell (@steveansell), Stephen B. Brown (@sbbrown), Jason Headley (@jasonheadley), Liam Steele (@pacificwhitesideddolphin), Stewart Wechsler (@stewartwechsler), and Peter L. Achuff (@plachuff) (https://inaturalist.ca/projects/bc-parks-inat-team-big-summer-2021?tab=identifiers). In addition, we appreciate the support from other members of the Reynolds and Starzomski labs for their creativity, input, and statistical advice on this project. Thank you especially to the BC Parks team of Sharilynn Wardrop, James Quayle, and Jen Grant; and the former Ministry of Forests, Lands, and Natural Resource Operations and Rural Development (Jennifer Psyllakis).

## Author Contributions

**Conceptualization:** Ellyne M. Geurts, John D. Reynolds, Brian M. Starzomski.

**Data curation:** Ellyne M. Geurts.

**Formal analysis:** Ellyne M. Geurts.

**Funding acquisition:** John D. Reynolds, Brian M. Starzomski.

**Investigation:** Ellyne M. Geurts.

**Methodology:** Ellyne M. Geurts, John D. Reynolds, Brian M. Starzomski.

**Project administration:** Ellyne M. Geurts, John D. Reynolds, Brian M. Starzomski.

**Resources:** John D. Reynolds, Brian M. Starzomski.

**Software:** Ellyne M. Geurts.

**Supervision:** John D. Reynolds, Brian M. Starzomski.

**Validation:** Ellyne M. Geurts, John D. Reynolds, Brian M. Starzomski.

**Visualization:** Ellyne M. Geurts.

**Writing – original draft:** Ellyne M. Geurts, John D. Reynolds, Brian M. Starzomski.

**Writing – review & editing:** Ellyne M. Geurts, John D. Reynolds, Brian M. Starzomski.

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
