## [Decision Letter · Decision Letter 0]

17 Apr 2023

PONE-D-23-09371

Not all who wander are lost: trail bias in community science

PLOS ONE

Dear Dr. Geurts,

Thank you for submitting your manuscript to PLOS ONE. After careful consideration, we feel that it has merit but does not fully meet PLOS ONE’s publication criteria as it currently stands. Therefore, we invite you to submit a revised version of the manuscript that addresses the points raised during the review process.

We look forward to receiving your revised manuscript.

Kind regards,

Dárius Pukenis Tubelis, Ph.D.

Academic Editor

PLOS ONE

Journal Requirements:

3. We note that Figures 1 and 4 in your submission contain copyrighted images. All PLOS content is published under the Creative Commons Attribution License (CC BY 4.0), which means that the manuscript, images, and Supporting Information files will be freely available online, and any third party is permitted to access, download, copy, distribute, and use these materials in any way, even commercially, with proper attribution. For more information, see our copyright guidelines: http://journals.plos.org/plosone/s/licenses-and-copyright.

a. You may seek permission from the original copyright holder of Figures 1 and 4 to publish the content specifically under the CC BY 4.0 license. 

Additional Editor Comments:

Dear Dr Ellyne Geurts,

Thank you for submitting your study about citizen science and trial bias to PLOS ONE.

Your submission has been evaluated by two reviewers, that suggested "Acceptance After Minor Revision".

Both reviewers consider that your manuscript is interesting, important and well-written. I agree with them.

Both reviewers were constructive and provided suggestions to improve the quality of your work. Thus, I ask

that you try to follow their suggestions, and my own coorections/suggestions provided below.

Reviewer 1 made several suggestions, and provided some references to improve your text.

You can add those that you consider appropriate.

This reviewer is mainly concerned with some questions in the methods, that you can easily improve/fix.

Reviewer 2 also made several suggestions, and provided some references. Your can insert those that you

consider suitable for your text. This reveiwer is mainly concerned with the sample based on the effort of only

"six observers". I ask that your research team read his/her comments carefully to better deal with this

question regarding representativeness (six in thousands of observers). Maybe you can mention the number of

observers in your studied region (substantially smaller than all iNaturalist observers). This reduced number of (six) observers would not be a big problem, but the reviewer´s concerns deserve attention, as there is research showing that observer´s characteristics influence the set of species documented by photographs. My suggestion is that you add some sentences in a separate paragraph of the Discussion to talk aobut this. Maybe, in a short section called "Shortcommings". It would be better if YOU deal with this in the Discussion, than if others write less soft criticism in the future.

Both reviewers also made several suggestions, questions and corrections on the manuscript (yellow marks).

Please try to follow those that you consider as appropriate.

Additional comments and corrections made by the editor:

Methods, by the end of page 5, line 100. You have to inform more clearly what types of organisms

were included in the surveys. You talk about species richness, exotic, native, but you do not explain

what organisms were studied (invertebrates, plants, vertebrates). You need an early sentence in the methods to clarify this.

Field experiments (line 128 ahead). How long were transects ? I could not find this information. Where is it ? Please clarify this aspect of transects. 

Through the whole text. Make sure that you use "observation" with the same meaning. It appears (but not really clear) that you considered "observation" as a photograph of a species that could be identified. Was it ? If so, please try to make a sentence for this. It is vague.

Please fix some problems in the References Section:

Ref 1. Title of the study/book: The initials of all words should not be in italics, except for the first word and names of regions, people, etc.

Please check this for all references.

Refs 3, 5 , others. For online articles, you should use "Available from:". You did not use "from". You did this mistake several times. Please fix.

Ref 6. It should not have a space beween initials of first and middle names. Also, delete the dot after initials. Please check this for all references.

Refs 7, 8, others. Some titles are divided into two parts, by using ":". When this occur, the first letter should not be in capital. Thus, use ": a case study" instead of ": A case study". It happened several times. Please check.

Ref 36. The scientific name of the species should be in italics. Please check all refs.

Reviewers' comments:

Reviewer's Responses to Questions

**Comments to the Author**

1. Is the manuscript technically sound, and do the data support the conclusions?

Reviewer #1: Yes

Reviewer #2: Yes

2. Has the statistical analysis been performed appropriately and rigorously? 

Reviewer #1: Yes

Reviewer #2: Yes

3. Have the authors made all data underlying the findings in their manuscript fully available?

Reviewer #1: Yes

Reviewer #2: Yes

4. Is the manuscript presented in an intelligible fashion and written in standard English?

Reviewer #1: Yes

Reviewer #2: Yes

5. Review Comments to the Author

Reviewer #1: The manuscript is well written and interesting. I have made some comments in the attached pdf. My major comment was trying to incorporate some results of a recently published ms (https://doi.org/10.1111/ibi.13169), with particular attention to their supplementary table about detection differences. I also suggested some other publications in the comments, marked some missing commas with yellow and suggested some clarifications in the methods.

Reviewer #2: The manuscript by Geurts et al provides an interesting bias evaluation on fine-scale of the contribution of citizen scientists on and off trails in Parks of British Columbia. The results shows that the public can provide a rich contribution observing the biodiversity without leaving the trails, a good perspective for the use of citizen-collected data. The text is well written and I only recommended some references to be cited in the introduction and some minor changes to improve the hole picture. See the pdf attached.

Actually, there is only one thing of my concern, which is related to the choice of using super-observers (only six) to run the field experiments. In fact, it is presumable that these observers are only a particular small subgroup representing volunteers of iNaturalist, because they likely have a superior capacity of organisms’ detection than most of other less productive observers. They are, at least, more experienced than most of other observers in the region to capture images from organisms and submit it to iNaturalist. It is true that the contribution of super-observers covers a large part of the dataset, but summed, the observations of most occasional observers use to be even larger. Other important issue is that they are only six observers. I know that it is a bit annoying, but volunteers may have different behaviors for surveying, often diverging their searching image, experience, taxonomic interest, and preferences (see: Bowler et al. 2022: https://doi.org/10.1038/s41598-022-15218-2; Boakes et al. 2016: https://doi.org/10.1038/srep33051; and Rosenblatt et al. 2022: https://doi.org/10.1093/ornithapp/duac008). For example, are these six observers somehow covering the main observers’ profile in the platform? Maybe it is not enough to cover either the variability inside the 43,690 observers from British Columbia. Considering that “the six” may not represent the most users of iNaturalist, in my opinion, this limitation should be highlighted in the text. Note that this issue does not decrease the relevance of the results and the quality of the contribution, but the inference deserves an adjustment.

Finally, I want to congratulate the authors for the manuscript and I hope to see this paper in the pages of PLOS ONE.

6. PLOS authors have the option to publish the peer review history of their article (what does this mean?). If published, this will include your full peer review and any attached files.

Reviewer #1: No

Reviewer #2: No

---

## [Author Response · Author response to Decision Letter 0]

19 May 2023

Our response exceeded 20,000 characters, so we uploaded our response as an attachment "Response to Reviewers" as directed in the email.

---

## [Editor Report · Decision Letter 1]

1 Jun 2023

Not all who wander are lost: trail bias in community science

PONE-D-23-09371R1

Dear Dr. Ellyne Geurts,

We’re pleased to inform you that your manuscript has been judged scientifically suitable for publication and will be formally accepted for publication once it meets all outstanding technical requirements.

Kind regards,

Dárius Pukenis Tubelis, Ph.D.

Academic Editor

PLOS ONE

Additional Editor Comments:

Dear Ellyne Geurts,

Thank you for submitting a corrected version of your manuscript about trial bias. I appreciated the changes and your responses to reviewers.

I consider that your submission can be accepted for publication in PLOS ONE.

There are some minor changes that I ask you to do during the next stages (likley during the proof correction).

Lines below refer to lines of your "clean" copy of the revised manuscript.

Line 90. Please replace "Figure 1" by "Fig 1".

Line 96. Fig 1. Please use bold for "a)", as done for other letters.

Line 142. Maybe you can inform here the lenght of the transects.....e.g. ...."paired xx m long transects". If variable, inform the max and min values. It sounds too vague. Bu it you want, leave as it is.

Lines 252-255. (Trail bias Section). The term "locally threatened" was repeated 4 times along 4 lines.

You could replace 2 of them by words such as "they" or "these taxa" or "among them"....

Line 272. Try to improve this sentence some way...."exotic species" is repeated twice.

Line 312. There is something wrong......maybe, you wanted to say: "....the opposite effect when forest coverage is low". Or ..."where"....

Dárius

---

## [Editor Report · Acceptance letter]

15 Jun 2023

PONE-D-23-09371R1 

Not all who wander are lost: trail bias in community science 

Dear Dr. Geurts:

I'm pleased to inform you that your manuscript has been deemed suitable for publication in PLOS ONE. Congratulations! Your manuscript is now with our production department. 

Kind regards, 

on behalf of

Dr. Dárius Pukenis Tubelis 

Academic Editor

PLOS ONE